# Self-reported illnesses in Thatta: Evidence from a rural and underdeveloped district in Sindh province, Pakistan

**Muhammad Ashar Malik**[1,2]*, **Rahat Batool**[1], **Muhammad Ahmed**[3], **Imran Naeem Abbasi**[1], **Zafar Ahmed Fatmi**[1], **Sarah Saleem**[1], **Sameen Siddiqui**[1]

1 Department of Community Health Sciences, Aga Khan University, Karachi, Pakistan, 2 Faculty of Arts and Sciences, Aga Khan University, Karachi, Pakistan, 3 NED University of Engineering and Technology, Karachi, Pakistan

* ashar.malik@aku.edu, ashar.malik@gmail.com

## Abstract

### Introduction

Self-reported illnesses (SRI) surveys are widely used as a low-cost substitute for weak Disease Surveillance Systems in low- and low-middle-income countries. In this paper, we report findings of a district-level disease prevalence survey of all types of illnesses including chronic, infectious, injuries and accidents, and maternal and child health in a rural district in Pakistan.

### Methods

A district-level survey was conducted in Thatta in 2019 with a population-representative sample of all ages (n = 7811) a. Survey included questions on demographics and SRIs from the respondents. Prevalence was estimated for all SRIs categorized into six major and 16 minor illnesses. The influence of important socio-demographic covariates on the illnesses and multiple comorbidities was explored by estimating prevalence ratios with a Generalized Linear Model of the Poisson family and by Zero-Inflated Poison Distribution respectively.

### Findings

36.57% of the respondents to the survey reported at least one SRI. Prevalence of communicable illnesses was 20.7%, followed by non-communicable illnesses (4.8%), Gastrointestinal disorders (4.4%), and injuries/disabilities (1.9%). Urban inhabitants were more likely to have Chronic Obstructive Pulmonary Disorders (3.34%) and Diabetes (1.62%). Females were most likely to have injuries (1.20,), disabilities (1.59), and Musculoskeletal Disorders (1.25). Children aged < 1 year (0.80) and elderly >65 years (0.78) were more likely to have comorbidities.

### Discussion

Our estimated prevalence of SRI is quite higher than the prevalence of unknown SRIs in national-level surveys in Pakistan. This research's findings serve as an example of aiding

**Data Availability Statement:** We cannot share data publicly because of legal and ethical restrictions. We have committed with the respondents of the survey at the time of singing

consent form on confidentiality of personal information that are not to be shared with any third party. Same commitment has been made in the ethical review application." However, we can consider individual request for accessing data by writing to zaheer.habib@aku.edu Senior Manager, Data Management Unit, Community Health Sciences Department, Aga Khan University.

**Funding:** This study was funded by the World Health Organization (WHO Registration Grant No.: 2018/824227-0; Purchase Order: 202084766). This grant was received by Dr. Sameen Siddiqui.

**Competing interests:** We (all authors) declared that we have no competing interests.

evidence-based priority settings within the health sector. Our findings on gender, and young and old age as positive predictors of SRI are consistent with similar surveys in a few LMICs.

## Recommendation and conclusion

We provide evidence of a complete disease profile of a district that is otherwise unavailable in the country. This study can reshape the existing health surveys and to aid evidence-based priority settings in the health sector. We, however, support strengthening the Disease Surveillance System as a reliable source of disease prevalence data.

## 1. Introduction

Reliable data on the prevalence of illnesses is the first line of evidence on priority settings within the health sector. Health ministers of a country routinely maintain such data in their respective Disease Surveillance System and Disease Early Warning System. However, in Low- and Low-Middle Income Countries (LMICs) including Pakistan such practices are weak and unreliable [1,2]. Reliance on intermittent population health surveys is a common alternative in such situations in many LMICs. However, such surveys are costly and often held with the help of financial assistance from development partners. These surveys focus on maternal and child health and this aspect limits their use for priority setting within the health sector.

Disease Surveillance System and Disease Early Warning System are weak and unreliable in Pakistan [2]. The common sources of prevalence data are level population-level surveys using the methodology of self-reported illnesses. However, population surveys lack data on all types of illnesses and multiple comorbidities, for example, USAID funded Demographic and Health Survey (DHS), and UNICEF funded Multi Indicator Cluster Survey (MICS) collect illness prevalence data on maternal and child health needs and their infectious illnesses [3,4]. World Bank-assisted Pakistan Living Standard Measurement Survey (PSLM) collects illness prevalence data on maternal, and childhood diarrhoea and unknown self-reported illnesses [5]. Such features of these surveys limit their use for priority settings in the health sector that require evidence on all types of illnesses. Few other types of surveys conducted in Pakistan are illness-specific and lack representative data, for example, the National Diabetes Survey of Pakistan 2016–17 or the Adult Tobacco Survey 2014 [6,7]. In the situation of limited evidence on disease prevalence, priority settings of resource allocation to the health sector and among healthcare needs usually follow expert opinion or historical budgeting with a strong influence of the development assistance favouring maternal and child health. For example, a study using financing data from 2000–2010 indicated that on average the share of the maternal and child health sector was 21% of the total funds allocated in 2010 to the health sector in the country with a substantial component of development assistance (51% of total Official Development Assistance to health sector) to Pakistan [8]. With the devolution of many functions of health to lower tiers of the government, the role of the federal Ministry of Health is limited in priority setting in the health sector. On the other hand, the lower tiers of the health sector cannot collect and maintain disease prevalence data.

In 2017 a collaboration was established between the academia and the health district administration to prototype evidence-based priority settings for the health department of Thatta District. A district-level disease prevalence survey was planned in 2019, as a stepping stone to overcome the challenges of priority settings on expert opinion and following historical patterns. In this paper, we report the findings of the population representative illness survey in

Thatta district. We provide prevalence estimates of all illnesses and their important socio-demographic covariates using the methodology of self-reported illnesses.

## 2. Methods

### 2.1 Survey settings

Thatta is an underdeveloped district in Sindh province in Pakistan. It is situated on the coast-line of the Arabian Sea in the province of Sindh, Pakistan. In the census of 2023, the population of Thatta District was 1.083 million [9]. The male population was 52.1% and the population density was 100/km$^2$. Over 80% of the population (approximately 0.98 million), lives in rural areas [9], and mainly relies on agriculture and fishing for their living. On the Human Development Index (HDI), it is ranked amongst the lowest: 90th out of 111 districts at the national level and 22nd among 23 districts in Sindh [10]. The health status and health-seeking patterns of the population are not so different from the HDI ranking of Thatta. For example, it was ranked among the highest in Sindh province in terms of under-five mortality (129 deaths per 10000 live births) and malnutrition (55%) and among the lowest for childhood vaccination (37% of children aged 1–2 years are fully immunized) [3,4]. Like the national situation, the ill-ness profile of district Thatta is limited to maternal and child health, immunization, and unknown common illnesses.

### 2.2 Survey design and data collection

The sample size was devised assuming a design effect of 1.5, a standard deviation (PKR 15,300) of demand for healthcare in rural Sindh, a margin of error of 10%, and a 10% refusal rate. The final sample size of the survey was 1,060 households. Multistage cluster sampling was used with a stratification strategy. This sample was distributed through sampling proportionate to populations among sub-districts/Talukas (Stratum) of Thatta namely 1) Thatta, 2) Mirpur Sakro, 3) Keti Bundar, 4) Ghorabari and 5) Kharo Chann: the latter two being managed jointly. Each sub-district was divided into rural and urban domains (Pakistan Bureau of Statistics, 2020). Rural and urban classification was carried out using the definition of rural union coun-cil and urban wards by the district administration of Thatta district. The primary sampling units were obtained from the smallest administrative unit of the district/local government, that is Union Councils in rural areas and Wards in urban areas. Three primary sampling units (vil-lages from Union Councils in rural areas and Mohallas/streets from Wards in urban areas) were selected at random in each UC/Ward. In each primary sampling unit, 8–12 households (secondary sampling units) were selected at random for data collection in the survey.

The survey questionnaire included demographic information and self-reported illness of every member of the household. Demographic data included age, gender, marital status, and schooling. Self-reported Illnesses included four categories: communicable illnesses (CIs) and non-communicable illnesses (NCIs), disabilities, and maternity and childbearing.

Data was collected by trained enumerators comprising a male and a female in each team. There were eight teams and a data collection supervisor. The data collectors were trained on data collection methods, cultural and religious sensitivities, the type and classification of ill-nesses, and the use of the computer tablet to enter data in the field. Data collection was com-pleted over four months, from January 2019 to April 2019. Data was collected from 1396 households (8635 individuals).

Informed consent was obtained from all respondents. An informed consent form translated into the Sindhi language was provided to the respondents and in the case of the illiterate respondents were read out to them. Four households refused to participate in the survey, while

73 respondents called the principal investigator on phone/cell numbers provided in the consent form and enquired about the survey. They were provided with all the details they needed.

Female and male heads of the households were interviewed by the male and female members of the data collection teams respectively. Both heads of the household provided demographic data of their household members including age, gender, marital status, schooling, and so forth. Following these interviews, data from household members over the age of 12 were collected. The enumerators visited each household twice. In the morning the female enumerator interviewed the females present in the household, while in the afternoon the male enumerator interviewed the male members. Data from the members under the age of 12 years was collected from at least one of their parents but preferably their mothers. Data on SRI was collected using a recall and record basis. Prevalence of CIs was based on the past month recall, while the prevalence of NCIs and disabilities was based on the presence of illnesses at the time of the survey. For maternity and childbearing, the recall period was within the preceding 12 months. The record component of SRI data collection included viewing prescriptions, diagnostic tests, and/or medicine invoices by enumerators.

Data validation was done during and after the completion of the data collection process. During the data collection, the supervisory team visited the field site and verified data collected from randomly picked respondents. After the data collection, a data validation exercise was carried out by a researcher from the University not involved in data collection. A random sample from the villages and wards was selected and ten households were contacted by telephone and were asked to reconfirm data collected on five randomly picked questions of the questionnaire.

During the data cleaning process, we dropped four households and 824 members. After dropping missing and incomplete data, the final sample for analysis is 1392 households (7811 members). This sample was higher than the calculated sample size of 1060 households.

The survey underwent ethical review by the Aga Khan University Ethical Review Board. After reviewing the application, they provided approval (letter number 2018-0615-836 on 24 November 2018).

## 2.3 Analysis

Demographic and socio-economic characteristics are reported in means and proportions. Survey sampling techniques are included in all the estimates. Age classifications of the World Health Organization were used to group the respondent by their age [11].

For the policy-relevant presentations of our findings, we reclassified four categories of SRI into six major categories (Communicable Illnesses, Mental Health and Non-Communicable illnesses, Gastrointestinal and Liver disorders, Injuries and Disabilities, Gynaecological and Obstetrics Disorders, and others/unclassified Disorders). Additionally, there are sixteen minor categories (Malaria and other Febrile illnesses, Upper respiratory tract infections, Common Infectious illnesses, Tuberculosis, Chronic obstructive pulmonary illnesses, Hypertension, Ischemic heart illness & Stroke, Diabetes, Mental disorders, Diarrhoea, Typhoid and other GI problems, Cirrhosis/Chronic liver illness/Hepatitis, Disabilities, Injury/Accident, Arthritis/Musculoskeletal disorders, Gynaecology and obstetrics and others/unclassified). The prevalence of illnesses was estimated as a proportion of respondents that reported an illness among all respondents of the survey. We obtained Confidence Intervals of prevalence by normal approximation. Prevalence ratios were estimated to account for crucial exposure variables such as gender (except in gynecological disorders), age, and least developed areas. We defined Gorabari and Keti Bandar as the least developed Talukas as these were ranked lowest on socio-economic indicators among the four Talukas in Thatta District [12].

We used prevalence ratios to estimate the prevalence of SRI in Thatta District following the validation study on relative benefits and harms of odd ratios and risk ratios and prevalence ratios in cross-sectional studies by Tamhane and Westfall et al (2016) and Coutinho and Sca-zufca et al (2008) [13,14]. The prevalence ratios were estimated using Generalized Linear Models of the Poison family [14]. Prevalence ratios are preferred over odd ratios to overcome the problem of overestimation and difficulties in the convergence of the model [14] though we acknowledge the limitation of PR of not satisfying the property of reciprocity o PR for ill versus PR of healthy [13].

To analyze factors influencing the multiple SRIs (0–4) we estimated the coefficient for each covariate using a Zero-inflated Poisson Distribution by a Bayesian marginal likelihood function with Laplace- Metropolis approximation [15]. All analyses, data cleaning, and imputation were carried out in STATA 15.1 while data were downloaded in Excel spreadsheets.

## 3. Findings

The final sample of this survey is 1392 households (7811 individuals). Females were 48% (n = 3710) of the sample. The adult population was 58% of the sample. Most of the adults were married (62%) and illiterate (75%). One-third (30%) of adults were employed at the time of the survey. Most of the population in Thatta district was rural (81%) except for Thatta Taluka where rural inhabitation was 67%. The average literacy rate was 23% in Thatta District. Among talukas, the proportion of females was (50.14%), and employed (31.02%) in Keti Bander, literate (32.89%) and living in urban areas (32.81%) in Thatta Taluka, married (64.2%) in Mirpur Sakro Taluka was higher than other talukas and the district averages (Table 1).

Nearly 37% of the respondents reported at least one SRI: 21% communicable illnesses, 5% non-communicable illnesses, 4% gastrointestinal and liver illnesses, 2% injuries and disabilities, and 2% other illnesses, while 3% of the women of reproductive age reported pregnancy-related health care needs. Among all SRIs, Malaria/fever and flu/cough were the most common illnesses reported by the respondents (10%) followed by upper respiratory tract infections (9.98%) (Table 2).

The estimated prevalence ratios revealed that being a female (PR 1.2, CIs 1.13–1.27), aged over 60 years (PR 1.54, CIs 1.41–1.69), and under five years (PR 1.42, CIs 1.33–1.52) are more likely to report an SRI. Living in urban areas (PR 1.51, CIs 1.42–1.61) and from least developed areas (PR 1.37, CIs 1.29–1.45) were more likely to report an SRI (Fig 1).

Generally being employed (PR 0.83, CIs 0.75–0.91) and living in large/extended families (PR 0.7, CIs 0.66–0.74) decreased the likelihood of reporting and SRI in Thatta district. The prevalence ratios of Diabetes (PR 7.78, CIs 4.71–12.84) and Arthritis/musculoskeletal disorders (PR 4.91, CIs 2.55–9.47) in over 60 years were among the highest (Table 3).

The socio-demographic factors that determined the multiple SRIs were similar to the factors determining a single SRI, except for the gender of the respondents. Being a female decreases the probability of multiple morbidities (Regression Mean -0.16, CIs -0.29- -0.04) in Thatta district. Living in an urban area increased the probability of multiple morbidities (Regression Mean 0.42, CIs 0.3–0.51) followed by living in the least developed areas (Regression Mean 0.29, CIs 0.15–0.41) whereas being currently employed and living in an extended family decreased the probability of multiple morbidities by 23% and 22% respectively (Table 4).

## 4. Discussions

In this study, we report the prevalence of illnesses including communicable illnesses, non-communicable illnesses, injuries/accidents, and maternity-related illnesses. To the best of our

**Table 1. Socio-economic and demographic profile of survey respondents.**

| Indicators | Talukas (Sub-districts) | | | | District Thatta |
|---|---|---|---|---|---|
| | Thatta | Mirpur Sakro | Ghorabari | Keti Bander | |
| **Full sample** | 42.07 | 33.57 | 15.32 | 9.04 | 7811 |
| **Gender** | | | | | |
| Male | 52.8 | 52.54 | 53.47 | 49.86 | 52.6 |
| Female | 47.2 | 47.46 | 46.53 | 50.14 | 47.5 |
| **Age** | | | | | |
| Less than 1 year | 3.59 | 3.33 | 3.43 | 3.69 | 3.48 |
| >1 and < = 5 | 13.3 | 14.15 | 13.47 | 14.49 | 13.73 |
| >5 and < = 16 | 28.67 | 30.3 | 31.97 | 29.4 | 29.81 |
| > 16 and < = 24 | 12.98 | 12.36 | 13.47 | 12.36 | 12.79 |
| >24 and < = 40 | 24.83 | 22.57 | 20.5 | 21.88 | 23.11 |
| >40 and < = 65 | 14.67 | 15.23 | 15.15 | 15.77 | 15.03 |
| 65 years and above | 1.97 | 2.07 | 2.01 | 2.41 | 2.05 |
| **Family size** | | | | | |
| 1–6 | 55.88 | 56.96 | 52.57 | 53.22 | 55.46 |
| 7+ | 44.12 | 43.04 | 47.43 | 46.78 | 44.54 |
| **Inhabitation** | | | | | |
| Rural | 67.19 | 89.7 | 96.07 | 86.67 | 80.93 |
| Urban | 32.81 | 10.3 | 3.93 | 13.33 | 19.07 |
| **Adults sample (+15 years)** | 43.56 | 32.80 | 14.72 | 8.92 | 4539 |
| **Marital Status** | | | | | |
| Married | 61.46 | 64.2 | 62.13 | 62.96 | 62.41 |
| Single/unmarried | 38.54 | 35.8 | 37.87 | 37.04 | 37.59 |
| **Literacy** | | | | | |
| can read and write | 32.89 | 16.37 | 13.66 | 17.37 | 23.26 |
| Can read | 1.73 | 0.47 | 0.45 | 0.25 | 0.99 |
| Can write | 1.02 | 0.2 | - | 0.25 | 0.53 |
| Neither can read nor can write | 64.37 | 82.95 | 85.89 | 82.13 | 75.22 |
| **Employment** | | | | | |
| Currently employed | 30.44 | 30.38 | 27.78 | 31.02 | 30.08 |
| Unemployed but seeking employment | 12.39 | 13.1 | 18.02 | 11.41 | 13.37 |
| Neither employed nor seeking employment | 57.16 | 56.52 | 54.2 | 57.57 | 56.55 |

The sample pertains to all respondents of the survey. The adult population pertains to respondents who were aged 15 years and above at the time of collection of data.

knowledge, the most recent such effort before our study is the National Health Survey which reported the complete disease profile of Pakistan in 1994–95 using burden of disease methodology [16]. We went a step ahead by explaining risk factors of the prevalence of illnesses and comorbidities that were not reported in national/provincial surveys including the National Health Survey 1994. Nevertheless, there are certain limitations to the interpretation of the results. Firstly, this survey was a rapid cross-sectional survey conducted in the spring and did not capture seasonal variation in the prevalence of illness. Secondly, findings on the prevalence of illness were validated with records, there was no clinical examination conducted during the data collection. Thirdly, we preferred PRs over ORs to overcome the problem of overestimation, but we acknowledge the limitations of PR that the property of reciprocity is not observed for PR for exposed/ill versus PR for unexposed/healthy [13]. Lastly, the estimated prevalence of a few illnesses is alarmingly low such as Diarrhoea, typhoid, and other GIs. One possible

**Table 2. Illness prevalence in Thatta.**

| Types of Disorders | No. | Prevalence |
|---|---|---|
| **Communicable Illnesses** | 1795 | 20.67% |
| | | 19.3–22.1 |
| Malaria and other febrile illnesses | 868 | 10.00% |
| | | 8.9%-11% |
| Upper respiratory tract infections | 867 | 9.98% |
| | | 8.9%-11% |
| Common Infectious illnesses | 36 | 0.41% |
| | | 0.2%-0.6% |
| Tuberculosis | 24 | 0.28% |
| | | 0.1%-0.5% |
| **NCI and Mental Health Disorders** | 414 | 4.77% |
| | | 4%-5.5% |
| Chronic obstructive pulmonary illnesses | 103 | 1.19% |
| | | 0.8%-1.6% |
| Hypertension | 108 | 1.24% |
| | | 0.9%-1.6% |
| Ischemic heart illness & Stroke | 85 | 0.98% |
| | | 0.6%-1.3% |
| Diabetes | 63 | 0.73% |
| | | 0.4%-1% |
| Mental disorders | 55 | 0.63% |
| | | 0.4%-0.9% |
| **Gastrointestinal and Liver Disorders** | 386 | 4.44% |
| | | 3.7%-5.2% |
| Diarrhoea, Typhoid, and other GI problems | 284 | 3.27% |
| | | 2.7%-3.9% |
| Cirrhosis/Chronic liver illness/Hepatitis | 102 | 1.17% |
| | | 0.8%-1.5% |
| **Injuries and Disabilities** | 166 | 1.91% |
| | | 1.4%-2.4% |
| Disabilities | 60 | 0.69% |
| | | 0.4%-1% |
| Injuries/Accidents | 55 | 0.63% |
| | | 0.4%-0.9% |
| Arthritis/ Musculoskeletal disorders | 51 | 0.59% |
| | | 0.3%-0.9% |
| **Gynaecology and obstetrics** | 278 | 3.20% |
| | | 2.6%-3.8% |
| **Other (unclassified) disorders** | 137 | 1.58% |
| | | 1.1%-2% |
| **Total** | 3176 | 36.57% |
| | | 39.1%-42.4% |

Prevalence is defined as the number of SRI (and by probing the respondents) in the respondents of the survey.

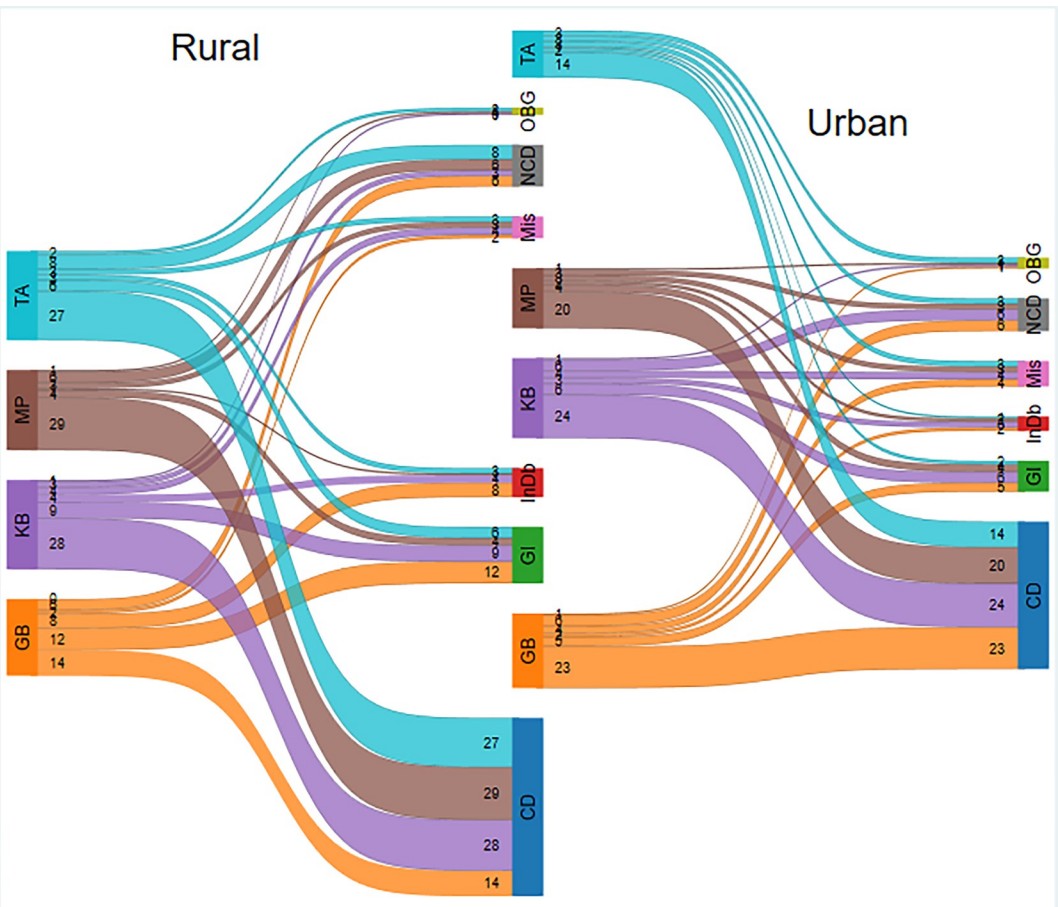

**Fig 1. Rural and urban prevalence of self-reported illnesses in sub-districts of Thatta.** Subdistricts GB: Ghorabari, KB: Keti Bunder, MP: Mirpur Sakro, TA: Thatta, Disease Classification CD: Communicable Disease, NCD: Non-Communicable Diseases Mental Health Disorders, GI: Gastrointestinal & Liver Disorders, InDb: Injuries & Disabilities, OBG: Gynaecology & obstetrics, Mis: Other disorders.

explanation for such low prevalence is that the survey was conducted in winter/spring. Another case is a low prevalence of NCI which could the to the lack of clinical examination for reporting an illness.

Estimates of the lone Burden of Disease study by the Pakistan Medical Research Council, in 1994 for NCIs (37.7%) and CI (38.4%) [16] were comparable, whereas in our case prevalence of CIs (20.67%) was higher than the prevalence of NCIs (6.68%, including injuries and disabilities 1.91%) in Thatta. These findings could partly be due to the illness classification used in our study and partly since our sample pertains to Thatta district which is the least developed district in Pakistan while the Burden of Disease study was drawn from a nationally representative sample. Our findings on the prevalence of SRIs (36.57%) in Thatta are higher than the national prevalence of unknown SRIs (7.38%) using a recall period of two weeks Pakistan Living Standard Measurement Surveys 2019–20 [5]. On the other hand, disability prevalence in our study is lower (0.7%) than disabilities reported in Thatta (3.01%) in PSLM 2019–20 [5]. We can speculate that such differences are due to differences of sample size, survey design effects, timing of survey, and recall methods.

We find few studies reporting the prevalence of illnesses using SRI approach and mainly LMICs from Latin America, Asia, and African contents, for example, Colombia [17], Vietnam

**Table 3. Illness prevalence ratios in Thatta in 2019.**

| Types of Disorders | Female | Over 60 years | Under 5 years | Literate | Current employed | Extended family | Urban | Least Developed Talukas |
|---|---|---|---|---|---|---|---|---|
| **Communicable Illnesses** | 0.91 | 0.81 | 1.98 | 0.72 | 0.68 | 0.64 | 1.47 | 1.28 |
| | 0.83–0.98 | 0.65–1.01 | 1.82–2.17 | 0.61–0.85 | 0.59–0.8 | 0.58–0.7 | 1.34–1.61 | 1.17–1.4 |
| Malaria and other febrile illnesses | 0.89 | 0.69 | 1.89 | 0.75 | 0.67 | 0.65 | 1.44 | 1.30 |
| | 0.78–1.02 | 0.48–0.99 | 1.65–2.18 | 0.58–0.95 | 0.53–0.84 | 0.57–0.75 | 1.24–1.66 | 1.13–1.5 |
| Upper respiratory tract infections | 0.93 | 0.78 | 2.20 | 0.69 | 0.70 | 0.61 | 1.51 | 1.20 |
| | 0.81–1.06 | 0.55–1.11 | 1.91–2.53 | 0.53–0.89 | 0.56–0.89 | 0.53–0.7 | 1.31–1.74 | 1.04–1.39 |
| Common Infectious illnesses | 0.82 | 1.97 | 0.97 | 1.29 | 0.38 | 0.61 | 1.14 | 2.28 |
| | 0.41–1.67 | 0.66–5.89 | 0.4–2.4 | 0.49–3.35 | 0.12–1.16 | 0.3–1.23 | 0.49–2.68 | 1.16–4.48 |
| Tuberculosis | 0.88 | 3.78 | 0.00 | 0.43 | 1.20 | 1.21 | 1.48 | 2.34 |
| | 0.38–2.05 | 1.39–10.28 | 0–0 | 0.09–1.96 | 0.39–3.63 | 0.52–2.86 | 0.55–3.99 | 1.08–5.07 |
| **NCI and Mental Health Disorders** | 1.13 | 4.18 | 0.50 | 1.64 | 1.04 | 0.84 | 2.00 | 1.83 |
| | 0.92–1.37 | 3.36–5.2 | 0.35–0.71 | 1.3–2.07 | 0.81–1.35 | 0.69–1.02 | 1.63–2.44 | 1.5–2.24 |
| Chronic obstructive pulmonary illnesses | 1.18 | 3.51 | 0.61 | 0.83 | 1.23 | 0.80 | 1.82 | 2.06 |
| | 0.78–1.78 | 2.12–5.78 | 0.33–1.13 | 0.45–1.51 | 0.71–2.11 | 0.54–1.19 | 1.19–2.78 | 1.39–3.04 |
| Hypertension | 1.89 | 4.62 | 0.07 | 2.53 | 1.28 | 0.97 | 3.11 | 1.63 |
| | 1.26–2.85 | 3.03–7.04 | 0.01–0.48 | 1.67–3.81 | 0.77–2.1 | 0.66–1.42 | 2.12–4.57 | 1.05–2.53 |
| Ischemic Heart illness & Stroke | 0.95 | 4.30 | 0.82 | 1.44 | 0.89 | 0.84 | 1.95 | 2.32 |
| | 0.6–1.5 | 2.53–7.32 | 0.43–1.57 | 0.83–2.5 | 0.48–1.65 | 0.54–1.28 | 1.22–3.13 | 1.5–3.58 |
| Diabetes | 0.76 | 7.78 | 0.14 | 2.28 | 1.24 | 0.75 | 1.17 | 1.38 |
| | 0.44–1.31 | 4.71–12.84 | 0.02–1.04 | 1.29–4.02 | 0.68–2.23 | 0.45–1.27 | 0.64–2.12 | 0.78–2.44 |
| Mental disorders | 0.83 | 1.17 | 0.67 | 1.15 | 0.49 | 0.81 | 1.64 | 1.62 |
| | 0.49–1.43 | 0.41–3.31 | 0.31–1.44 | 0.5–2.62 | 0.19–1.27 | 0.48–1.39 | 0.92–2.92 | 0.93–2.8 |
| **Gastrointestinal and Liver Disorders** | 1.55 | 2.26 | 1.52 | 1.23 | 1.46 | 0.80 | 1.80 | 1.81 |
| | 1.25–1.93 | 1.65–3.09 | 1.19–1.95 | 0.91–1.66 | 1.07–1.99 | 0.65–0.98 | 1.44–2.26 | 1.46–2.24 |
| Diarrhoea, Typhoid, and other GI problems | 1.54 | 2.02 | 2.17 | 1.33 | 1.36 | 0.85 | 1.91 | 1.77 |
| | 1.19–1.98 | 1.35–3.03 | 1.66–2.85 | 0.93–1.92 | 0.93–1.98 | 0.67–1.08 | 1.47–2.48 | 1.37–2.28 |
| Cirrhosis/Chronic liver illness/Hepatitis | 1.60 | 2.74 | 0.17 | 1.04 | 1.68 | 0.67 | 1.50 | 1.91 |
| | 1.02–2.51 | 1.61–4.66 | 0.05–0.53 | 0.59–1.83 | 0.94–2.98 | 0.44–1.01 | 0.93–2.42 | 1.25–2.92 |
| **Injuries and Disabilities** | 0.98 | 2.47 | 0.54 | 0.54 | 0.96 | 0.98 | 2.16 | 1.81 |
| | 0.71–1.35 | 1.57–3.87 | 0.33–0.88 | 0.31–0.93 | 0.6–1.53 | 0.72–1.32 | 1.54–3.03 | 1.29–2.53 |
| Disabilities | 0.59 | 1.37 | 0.72 | 0.55 | 0.47 | 1.14 | 1.45 | 0.69 |
| | 0.35–1 | 0.53–3.5 | 0.36–1.45 | 0.21–1.4 | 0.19–1.15 | 0.69–1.87 | 0.82–2.57 | 0.35–1.35 |

(*Continued*)

**Table 3.** (Continued)

| Types of Disorders | Female | Over 60 years | Under 5 years | Literate | Current employed | Extended family | Urban | Least Developed Talukas |
|---|---|---|---|---|---|---|---|---|
| Injury/Accident | 1.31 | 1.82 | 0.80 | 0.45 | 1.68 | 1.12 | 1.94 | 2.99 |
|  | 0.76–2.27 | 0.71–4.65 | 0.38–1.72 | 0.16–1.28 | 0.81–3.52 | 0.66–1.9 | 1.04–3.63 | 1.72–5.21 |
| Arthritis/ Musculoskeletal disorders | 1.44 | 4.91 | 0.00 | 0.64 | 1.18 | 0.70 | 3.89 | 2.81 |
|  | 0.74–2.78 | 2.55–9.47 | 0–0 | 0.27–1.52 | 0.5–2.82 | 0.39–1.26 | 2.15–7.02 | 1.53–5.16 |
| **Gynaecology and obstetrics** | - | - | - | 1.14 | 0.18 | 0.65 | 0.97 | 1.20 |
|  | - | - | - | 0.78–1.66 | 0.09–0.34 | 0.51–0.83 | 0.72–1.3 | 0.93–1.55 |
| **Other (unclassified) disorders** | 1.190 | 4.663 | 1.324 | 0.876 | 0.997 | 0.839 | 0.966 | 0.460 |
|  | 0.83–1.7 | 3.1–7.01 | 0.83–2.11 | 0.51–1.49 | 0.59–1.68 | 0.59–1.2 | 0.63–1.49 | 0.27–0.77 |
| **Being Ill** | 1.20 | 1.54 | 1.42 | 0.95 | 0.83 | 0.70 | 1.51 | 1.37 |
|  | 1.13–1.27 | 1.41–1.69 | 1.33–1.51 | 0.87–1.04 | 0.75–0.91 | 0.66–0.74 | 1.42–1.61 | 1.29–1.45 |

Prevalence ratios are obtained as exponentiated coefficients (95% confidence intervals in parenthesis) of a generalized linear model for the Poisson family with a logarithmic link function. The exposure variables include being a female (except Gynaecology and obstetrics), age categories, being literate (can read and write), living in an extended family, living in urban areas, and living in least developed talukas (Kati-Bandar or Ghorabari).

**Table 4. Determinants of multiple morbidities (0–4) in Thatta in 2019.**

| Determinants | Mean (95% Equal tailed Creditable Intervals) |
|---|---|
| Female | -0.16 |
|  | -0.29- -0.04 |
| Under 5 years | 0.18 |
|  | 0.05–0.35 |
| Over 60 years | 0.13 |
|  | -0.03–0.27 |
| Urban | 0.42 |
|  | 0.3–0.51 |
| Literate | -0.170 |
|  | -0.43–0.06 |
| Currently employed | -0.23 |
|  | -0.36–0.1 |
| Extended family | -0.22 |
|  | -0.32–0.11 |
| Least developed Talukas | 0.29 |
|  | 0.15–0.41 |
| BIC | 12342.34 |
| Sample | 4254 |

Sample pertains to respondents of self-reported illnesses: Healthy and or having 1–4 illnesses. Estimates are obtained with a Bayesian zero-inflated Poisson regression using a Marginal likelihood (ML) by Laplace-Metropolis approximation. BIC is the Bayesian information criterion.

[18,19], Botswana [20], Nepal [21], Bangladesh [22], Myanmar [23], Uganda [24] and Cambodia [25]. A possible explanation for the popularity of SRI surveys is weak disease surveillance systems in many LMICs [26]. Moreover, population health surveys are expensive and conducted with financial assistance from development partners and follow their priorities, often restricted to infectious illnesses and maternal and child health such as Living Standard Surveys sponsored by the World Bank, Multi-Indicator Cluster Surveys by UNICEF, and Demographic and Health Surveys by the USAID [3–5]. However, using the common methodology in these surveys enables comparison across countries while in the case of SRI surveys, the methods varied across countries for the type of illnesses, recall period, and geographical focus making the comparison of results challenging. For example, some SRI surveys were carried out on a small scale covering all types of illnesses in Vietnam, Cambodia, Bangladesh, and Nepal [18,21,22,25]. Few studies of SRIs used common recall for NCIs and CIs [18,21,22,25] and acute/communicable illnesses survey by SeoAung and MyintOo et al (2015) used a 90 days recall period [23]. Illness surveys that focused on NCIs enquired about illnesses based on "ever diagnosed/ informed by a physician or health worker" [17,19,20,24,27]. Except for Rehman and Gilmour et al, (2013), the SRI surveys included all illnesses and were conducted in rural areas [18,21,25]. Surveys on NCI, on the other hand, were carried out in urban areas [23] or were conducted at a larger scale [17,20,27].

Estimates of SRI in this paper (36.6%) are lower than the estimates of SRI in Bangladesh [22] (45%, n = 1593 households), and SRIs estimates in Vietnam (47.7%, n = 48919), but higher than SRIs estimates in Cambodia (15.05%, n = 33161) (Ir and Men,2010) and in Nepal (24.5%, n = 6580) [19,21,25]. These studies used a rapid data collection: 3–4 months and focused on all ages and all illnesses. However, the recall period in the case of Giang and Allebeck (2003), Rehman and Gilmour (2013), and Paudel (2020) was the previous four weeks, while in the case of Ir and Men (2010), the recall period was the previous one year [18,21,22,25]. Moreover, in the case of Ir and Men (2010) the SRI data was collected by trained data collectors and was verified by a public health doctor while in the case of Giang and Allebeck (2003), Rehman and Gilmour (2013) such steps of enhancing quality of data collection were missing [18,22,25]. Such variation in survey design may have influenced the prevalence of an illness. For example, diabetes prevalence was found to be as low as 1.1% in Uganda, 5.7% in Colombia, and 9.3% (metabolic illnesses) in India [17,24,28]. While in Colombia and India, the sample size was large: 11 districts and country level respectively, in Uganda, the study was carried out in an urban district [17,24,28]. The period of data collection spanned over one year in Colombia and India [17,28]. In Uganda and Colombia, the sample was drawn from the adult population, and in India, the sample included all ages [24,28].

Our findings on the risk factors (age, gender, and residential status) of SRI or comorbidities with literature included in this paper. Our finding that women are more likely to report illness (es) and multiple morbidities is consistent with findings from Colombia, Vietnam, Cambodia, Nepal, Myanmar, Botswana, India, Bangladesh, and Uganda [17,20–25,28].

Our findings on the role of old age (aged 60 years and above) reporting SRIs or NCIs, is consistent with findings from Colombia, Myanmar, Botswana, Bangladesh, and Vietnam [17,18,20,22,23].

Our findings that living in urban areas increases the prevalence of SRIs over living in rural areas are consistent with similar findings from Colombia, Botswana, and India [17,20,28]. On unemployment as a predictor of comorbidities, our findings are similar to findings from Vietnam: higher likelihood of NCIs for unemployed than employed (OR 1.59, CIs 0.96–2.69) [19] whereas on household size our findings are similar to findings from Bangladesh (OR 0.89, CIs 0.82–0.87) and India reporting a decreasing proportion of illnesses with increasing family size (<4 members 26.8, 5–8 members 26.0% and >9 members 20.2%) [22,28].

## 5. Recommendations and conclusion

For multiple reasons, our estimates are better than the existing evidence of illness prevalence in Pakistan. We recommend that the policymakers advocate for replacing unknown self-reported illnesses with descriptions of all types of illnesses in the PSLM survey. Similarly, district health administrations in other provinces in Pakistan can replicate our survey for evidence-based decision-making within their respective districts. However, we conclude that strengthening the Disease Surveillance System and Disease Early Warning System are crucial elements for evidence-based priority settings in the health sector at national and sub-national levels in Pakistan.

## Acknowledgments

We gratefully acknowledge the survey's field staff for their contributions and the district health office officials, Thatta, for their logistical support.

## Author Contributions

**Conceptualization:** Muhammad Ashar Malik, Sameen Siddiqui.

**Data curation:** Muhammad Ashar Malik.

**Formal analysis:** Muhammad Ashar Malik, Rahat Batool.

**Investigation:** Muhammad Ashar Malik, Muhammad Ahmed, Zafar Ahmed Fatmi.

**Methodology:** Muhammad Ashar Malik, Zafar Ahmed Fatmi, Sarah Saleem, Sameen Siddiqui.

**Project administration:** Muhammad Ashar Malik.

**Supervision:** Muhammad Ashar Malik, Sarah Saleem.

**Validation:** Muhammad Ashar Malik.

**Visualization:** Muhammad Ashar Malik, Muhammad Ahmed, Imran Naeem Abbasi.

**Writing – original draft:** Muhammad Ashar Malik, Rahat Batool, Sameen Siddiqui.

**Writing – review & editing:** Muhammad Ashar Malik, Muhammad Ahmed, Imran Naeem Abbasi, Zafar Ahmed Fatmi, Sarah Saleem, Sameen Siddiqui.

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
