## [Decision Letter · Decision Letter 0]

12 Dec 2023

PONE-D-23-33013Self-reported illnesses in Thatta: Evidence from a rural and underdeveloped district in Sindh province, PakistanPLOS ONE

Dear Dr. Malik,

Thank you for submitting your manuscript to PLOS ONE. After careful consideration, we feel that it has merit but does not fully meet PLOS ONE’s publication criteria as it currently stands. Therefore, we invite you to submit a revised version of the manuscript that addresses the points raised during the review process. Detailed comments, by reviewers and editors are appended below. We look forward to receiving your revised manuscript.

We look forward to receiving your revised manuscript.

Kind regards,

Adnan A. Khan

Academic Editor

PLOS ONE

“We gratefully acknowledge the World Health Organization, Regional Office, Eastern Mediterranean Region for providing funding for this survey. Our acknowledgments are due to the field staff of the survey for their contributions to the survey and to the officials of the district health office, Thatta for their logistic support.”

“World Health Organization regional office of the Eastern Mediterranean Region supported this survey”

6. Please amend your manuscript to include your abstract after the title page.

7. Please include a separate caption for each figure in your manuscript.

8. We note that Figure 1 in your submission contain [map/satellite] images which may be copyrighted. All PLOS content is published under the Creative Commons Attribution License (CC BY 4.0), which means that the manuscript, images, and Supporting Information files will be freely available online, and any third party is permitted to access, download, copy, distribute, and use these materials in any way, even commercially, with proper attribution. For these reasons, we cannot publish previously copyrighted maps or satellite images created using proprietary data, such as Google software (Google Maps, Street View, and Earth). For more information, see our copyright guidelines: http://journals.plos.org/plosone/s/licenses-and-copyright.

9. Please include your tables as part of your main manuscript and remove the individual files. Please note that supplementary tables (should remain/ be uploaded) as separate "supporting information" files

Additional Editor Comments:

This is an interesting study which has potential to inform healthcare utilization. However, as pointed out by reviewers, major and substantial revisions are needed before it can be accepted for publication. In addition to specific comments, including those in the manuscripts by Reviewer 2, the authors may consider the following:

While the authors suggest the use of SRI to direct priorities in health systems, essentially an exercise in burden of disease estimation, the data presented does not support this assertion. The term "burden of disease" is repeatedly mentioned. Instead it would be more accurate to use this data to understand what communities are feeling in terms of symptoms/ disease self perceptions and use this information to improve healthcare provision. If this was the intent, please carefully re word to communicate this.

Some claims such as "our estimates of burden of disease are much higher and comprehensive than the current evidence of health seeking in Thatta" (Discussion section in abstract), need to be backed with evidence. What are the study findings being compared against. Are there BoD or healthcare seeking estimates from Thatta available. Is it the same or different in public and private sector?

In the introduction section, there is a need to elaborate why SRI framework was used, its applicability and limitations to the context of the study. Mention of low capacity by district adminstrators needs to be both elaborated upon and also corroborated with evidence

There is a need to copy edit final version before submission to avoid simple grammar and word ommission errors

Methods:

The survey and sampling design is well described.

It must be specified how many interviews were conducted per HH, specifically, if the husband and wife were both interviewed or one interviewee per HH was selected. If latter there should be analysis of the effect of this designation on the results

It should also be highlighted that answers for children are from their parents and therefore not "self reported".

Verban autopsy is mentioned in the discussion as a means to verify self reports. The methodology used is not mention. There is no mention of verbal autopsy in results either

Line 90 mentions two methods of verification were used. Please specify #2. Were any statistical means used to measure verification?

Assumptions of prevalence ratio methodology used may be discussed

Results:

Table 2 suggests that the prevalences are for the entire year, please confirm. The period for which self reported illnesses were asked for is not specified

Table 2 describes prevalences. Some of these vary widely (log order differences) than national figures. For example, Hypertension of 1.24%, Diabetes: 0.73% compare with national figures of 30-50% and 17-26% respectively. There is a need to discuss these differences and their posited rationale. Diarrhea is reported at 3.27% for the year, when PSLM data suggests around 16% in 2 weeks prior to survey

Table 4 may benefit from the use of continuous age variable. Otherwise there is concern that some of the lower morbidity among women may actually be a cohort effect (women are often younger than men in surveys, whether this is true or not must be mention)

Verbal autopsy (mentioned in discussion, line 159) is not mentioned in results

Discussion:

There needs be a discussion of log order lower prevalence of many common conditions in this survey compared to the national samples. The assertion that prevalences are "alarmingly high" (line 154) is not corroborated by national data

While the discussion of SRI literature (likely better placed in introduction) is extensive, it needs to clarify what is intended with this review and how this review serves this study

SRI data presented has not been validated and can not be used to estimate about burden of disease

Unclear what is meant with unknow SRI (lines 199-201)

While there is extensive citing of literature in discussion, it may be supplemented by interpreting the findings AND THEN connect this interpretation to the cited literature

In Recommendations, it is suggested that a combined SRI type survey may substitute for individual disease survey. This is not justified based on the data presented. If anything the data suggests that SRI may massively under identify serious conditions

Reviewers' comments:

Reviewer's Responses to Questions

**Comments to the Author**

1. Is the manuscript technically sound, and do the data support the conclusions?

Reviewer #1: Partly

Reviewer #2: Partly

Reviewer #3: Yes

2. Has the statistical analysis been performed appropriately and rigorously? 

Reviewer #1: No

Reviewer #2: Yes

Reviewer #3: Yes

3. Have the authors made all data underlying the findings in their manuscript fully available?

Reviewer #1: No

Reviewer #2: No

Reviewer #3: No

4. Is the manuscript presented in an intelligible fashion and written in standard English?

Reviewer #1: No

Reviewer #2: No

Reviewer #3: Yes

5. Review Comments to the Author

Reviewer #1: This research titled as "Self-reported illnesses in Thatta: Evidence from a rural and underdeveloped district in Sindh province, Pakistan" lacks a strong conceptual foundation. The study's premise relies on a superficial analysis of existing literature, failing to adequately highlight any discernible literature gap. An effective study typically builds upon existing research by identifying gaps or unexplored areas, contributing valuable insights to the field. However, regrettably, this manuscript appears to fall short in this regard.

The absence of a clear identification of the literature gap significantly diminishes the scholarly impact of this work. A comprehensive understanding of existing research is fundamental to conducting a meaningful study, enabling the identification of areas where new knowledge can be generated. Unfortunately, this manuscript fails to provide such a distinctive contribution to the established body of knowledge in its respective field.

For a study to be considered robust and impactful, it should not only engage with relevant literature but also delineate how it extends or challenges existing knowledge. This manuscript, regrettably, does not adequately fulfill this criterion. Hence, it may not significantly advance the field or offer novel perspectives, rendering it less impactful in contributing to the discourse within the academic realm.

Reviewer #2: I have completed a thorough review of your manuscript titled "Self-reported illnesses in Thatta: Evidence from a rural and underdeveloped district in Sindh province, Pakistan." I commend the effort you have invested in your research, and I recognize the importance of the manuscript's topic. However, I must convey that, upon careful examination, the manuscript requires significant revisions before it can advance further in the processing stage.

The manuscript faces challenges in terms of clarity in several sections, making it difficult for readers to seamlessly follow the content. I suggest considering a restructuring of sentences and paragraphs to enhance overall coherence.

Numerous grammatical errors and language issues were identified throughout the manuscript. To improve the overall quality of the writing, I recommend a thorough language editing process.

The references require careful attention, as some instances deviate from the journal's prescribed reference guidelines. It is essential to ensure adherence to the specified format for all references.

Furthermore, a comprehensive and well-articulated conclusion is currently lacking. I encourage you to summarize your key findings and discuss their implications, providing a strong closing statement for the manuscript.

Additionally, it is important to note that I could not review the tables in their entirety, as they were not present in the submitted file. I kindly request that you ensure the inclusion of tables in your revised submission, enabling a comprehensive review of all elements of the manuscript.

Your attention to these revisions will significantly contribute to the overall quality and readiness of your manuscript for further processing. Please see the attached manuscript for my detailed comments.

Reviewer #3: It would be good to reviewdata tables for easier understanding of key findings. Tables were not available in the manuscript, however, reference to tables is given. Maybe there is an error in uploading / downloading.

Tone of some sentences may be corrected, e.g. rather than "illetrate" it would be better to use the word un-educated.

Since the study focuses on just one district, it should be mentioned as a limitation of the study.

6. PLOS authors have the option to publish the peer review history of their article (what does this mean?). If published, this will include your full peer review and any attached files.

Reviewer #1: No

Reviewer #2: **Yes: **Imran Hameed Khaliq

Reviewer #3: **Yes: **Naeem Majeed

---

## [Author Response · Author response to Decision Letter 0]

25 Jul 2024

We have provided details of participants consent and data collection process in details in the methods section of the revised manuscript.

The grant information is appended below and have been added in both sections

The details of WHO Registration Grant No.: 2018/824227-0, Purchase Order: 202084766, Amount: USD 50K.

“We gratefully acknowledge the World Health Organization, Regional Office, Eastern Mediterranean Region for providing funding for this survey. Our acknowledgments are due to the field staff of the survey for their contributions to the survey and to the officials of the district health office, Thatta for their logistic support.”

“World Health Organization regional office of the Eastern Mediterranean Region supported this survey”

Reply: The acknowledgement section is edited according to the editor’s advice.

Reply: We cannot share data publicly because of legal and ethical restrictions. We have committed with the respondents of the survey at the time of singing consent form on confidentiality of personal information that are not to be shared with any third party. Same commitment has been made in the ethical review application." However, we can consider individual request for accessing data by writing to zaheer.habib@aku.edu Senior Manager, Data Management Unit, Community Health Sciences Department, Aga Khan University. 

6. Please amend your manuscript to include your abstract after the title page.

Reply: Abstract is appended after the title page.

7. Please include a separate caption for each figure in your manuscript.

8. We note that Figure 1 in your submission contains [map/satellite] images which may be copyrighted. All PLOS content is published under the Creative Commons Attribution License (CC BY 4.0), which means that the manuscript, images, and Supporting Information files will be freely available online, and any third party is permitted to access, download, copy, distribute, and use these materials in any way, even commercially, with proper attribution. For these reasons, we cannot publish previously copyrighted maps or satellite images created using proprietary data, such as Google software (Google Maps, Street View, and Earth). For more information, see our copyright guidelines: http://journals.plos.org/plosone/s/licenses-and-copyright.

We used ArcMap software version10.8 licensed to NED University, Karachi, Pakistan to develop figure 1. The data utilized in this figure is not sourced from proprietary sources (whatsoever) such as Google Maps, Street View, or Earth. 

This statement is now appended in the notes below figure 1

9. Please include your tables as part of your main manuscript and remove the individual files. Please note that supplementary tables (should remain/ be uploaded) as separate "supporting information" files

We have included tables in the main text.

 Additional Editor Comments:

This is an interesting study which has potential to inform healthcare utilization. However, as pointed out by reviewers, major and substantial revisions are needed before it can be accepted for publication. In addition to specific comments, including those in the manuscripts by Reviewer 2, the authors may consider the following:

While the authors suggest the use of SRI to direct priorities in health systems, essentially an exercise in burden of disease estimation, the data presented does not support this assertion. The term "burden of disease" is repeatedly mentioned. Instead, it would be more accurate to use this data to understand what communities are feeling in terms of symptoms/ disease self perceptions and use this information to improve healthcare provision. If this was the intent, please carefully re word to communicate this.

We have removed the term burden of disease from the revised version of the manuscript and consistently used the term self-reported illnesses.

Some claims such as "our estimates of burden of disease are much higher and comprehensive than the current evidence of health seeking in Thatta" (Discussion section in abstract), need to be backed with evidence. What are the study findings being compared against. Are there BoD or healthcare seeking estimates from Thatta available. Is it the same or different in public and private sector?

We have added estimated prevalence of certain illnesses reported in national and provincial surveys to clarify our comparison.

In the introduction section, there is a need to elaborate why SRI framework was used, its applicability and limitations to the context of the study. Mention of low capacity by district adminstrators needs to be both elaborated upon and corroborated with evidence.

We have provided limitation of SRI. We have provided reference to our claims of low capacity of district administrators. 

There is a need to copy edit final version before submission to avoid simple grammar and word omission errors.

We have reviewed our revised manuscript by an native English speaking scholar. 

Methods:

The survey and sampling design is well described.

It must be specified how many interviews were conducted per HH, specifically, if the husband and wife were both interviewed or one interviewee per HH was selected. If latter there should be analysis of the effect of this designation on the results

It should also be highlighted that answers for children are from their parents and therefore not "self-reported".

We mentioned the process of data collection from children below 2 years of age in paragraph 3 of Survey Design and Data Collection.

Verban autopsy is mentioned in the discussion as a means to verify self-reports. The methodology used is not mentioned. There is no mention of verbal autopsy in results either.

We removed the term “verbal autopsy” and provided more details on methods of collection data from the respondents. 

Line 90 mentions two methods of verification were used. Please specify #2. Were any statistical means used to measure verification?

We specified two methods of verification, during the data collection and after the data collection in the revised manuscript in the second last paragraph of section on Survey Design and Data Collection.

Assumptions of prevalence ratio methodology used may be discussed.

We provided limitations of prevalence ratios in the first paragraph of the discussion section and cited some literature in support of it. However, this is not a study on validating methods of reporting risk factors/ covariates of SRIs, but we can redo the analysis using odd ratios if the reviewer may wish.

Results:

Table 2 suggests that the prevalences are for the entire year, please confirm. The period for which self-reported illnesses were asked for is not specified.

We have provided the period of data collection in paragraph 3 of section on data collection. We also corrected the heading of table 2.

Table 2 describes prevalences. Some of these vary widely (log order differences) than national figures. For example, Hypertension of 1.24%, Diabetes: 0.73% compared with national figures of 30-50% and 17-26% respectively. There is a need to discuss these differences and their posited rationale. Diarrhea is reported at 3.27% for the year, when PSLM data suggests around 16% in 2 weeks prior to survey.

We have provided such comparisons in the discussion section and the challenges making such comparisons. 

Table 4 may benefit from the use of continuous age variable. Otherwise, there is concern that some of the lower morbidity among women may be a cohort effect (women are often younger than men in surveys, whether this is true or not must be mention)

The mean age difference between males in females was 0.34 years so we don’t anticipate such bias introduced in the model. Moreover, we included >5 years and<60 years purposefully learning from literature about greater healthcare need such as Malik and Azam, 2012. (Muhammad Malik, A., & Azam Syed, S. I. (2012). Socio-economic determinants of household out-of-pocket payments on healthcare in Pakistan. International journal for equity in health, 11, 1-7.)

Verbal autopsy (mentioned in discussion, line 159) is not mentioned in results.

We have removed the term verbal autopsy from the methods section and have provided more details in the section on methods of data collection.

Discussion:

There needs be a discussion of log order lower prevalence of many common conditions in this survey compared to the national samples. The assertion that prevalences are "alarmingly high" (line 154) is not corroborated by national data

We have edited this argument that comparison with national survey is not possible because of difference in age cohort focused in these surveys. 

While the discussion of SRI literature (likely better placed in introduction) is extensive, it needs to clarify what is intended with this review and how this review serves this study

The objective of this review is to advocate the challenges with comparison of estimates of SRIs due to difference in survey design, period, geographical coverage, methods of analysis and data collection.

SRI data presented has not been validated and cannot be used to estimate about burden of disease

We mentioned this limitation in the revised manuscript.

Unclear what is meant with unknow SRI (lines 199-201)

We replaced “unknown” with “other” SRI

While there is extensive citing of literature in discussion, it may be supplemented by interpreting the findings AND THEN connect this interpretation to the cited literature

In Recommendations, it is suggested that a combined SRI type survey may substitute for individual disease survey. This is not justified based on the data presented. If anything, the data suggests that SRI may be massively under identify serious conditions.

In the revised manuscript, we did provide interpretation of our results from other sources in Pakistan and next we compare our findings with literature outside Pakistan. 

We 

---

## [Editor Report · Decision Letter 1]

11 Sep 2024

PONE-D-23-33013R1Self-reported illnesses in Thatta: Evidence from a rural and underdeveloped district in Sindh province, PakistanPLOS ONE

Dear Dr. Malik,

Thank you for submitting your manuscript to PLOS ONE. After careful consideration, we feel that it has merit but does not fully meet PLOS ONE’s publication criteria as it currently stands. Therefore, we invite you to submit a revised version of the manuscript that addresses the points raised during the review process.

We look forward to receiving your revised manuscript.

Kind regards,

Adnan Ahmad Khan

Academic Editor

PLOS ONE

Additional Editor Comments:

Thank you for the revisions that have clarified some of the measurement and findings. I would like to draw your attention to the Introduction and Discussion sections and ask for clarity in what is being conveyed. Following are some issues to address:

1. In the Introduction section, it is unclear what is the point of this study. Is it intended as an alternative to current BoD surveys and modeling? If so a connection must be made to BoD literature, confluence points and why this study would add to the knowledge base

2. In the Introduction section, it is unclear what is the point of Para on pg 4 from line 81

3. In the Findings section, it appears that for acute/transient illnesses, only the past two weeks period is included. This would mean different things if extrapolated for an entire year or to the community as a measure of burden of SRI. Please clarify what is the denominator and if you are intending a point in time prevalence or is it for the year - and what adjustments/ analyses would be needed for each

4. In the Discussion section, the first sentence is unclear. What example does the study set. It should be clarified what is the value of this information and how does it enhance the knowledge in the field. Ideally one would like to see an framework where findings of this study can be linked to BoD. Also, the discussion section should open by positing what the study has shown and what salient findings would be discussed.

5. The subsequent paras in the Discussion attempt to explain the findings in the schema: SRI, then NCI, but the discussion goes back and forth with circling back to points. Better organization and flow of logic would help understand the message.

6. Some parts of the narrative are unclear. For e.g., the sentence on Line 307 in unclear. What could not be synthesized. Prevalence with measures of variation should be available in the data collected. Is there a desire to compare these to literature. This is not clear.

7. Better overall structuring of the narrative in Intro and Discussions with attention to flow of logic would help

The study has potential for an important contribution to the body of knowledge on the subject, but would require substantial re-writing

---

## [Author Response · Author response to Decision Letter 1]

8 Nov 2024

Response to reviewer is provided in the attached file, please.

---

## [Editor Report · Decision Letter 2]

25 Nov 2024

Self-reported illnesses in Thatta: Evidence from a rural and underdeveloped district in Sindh province, Pakistan

PONE-D-23-33013R2

Dear Dr. Malik,

We’re pleased to inform you that your manuscript has been judged scientifically suitable for publication and will be formally accepted for publication once it meets all outstanding technical requirements.

Kind regards,

Adnan Ahmad Khan

Academic Editor

PLOS ONE
---

## [Editor Report · Acceptance letter]

17 Dec 2024

PONE-D-23-33013R2 

PLOS ONE

Dear Dr. Malik, 

I'm pleased to inform you that your manuscript has been deemed suitable for publication in PLOS ONE. Congratulations! Your manuscript is now being handed over to our production team.

Kind regards, 

on behalf of

Dr Adnan Ahmad Khan 

Academic Editor

PLOS ONE